

# Accounting for Hydroclimatic Properties in Flood Frequency Analysis Procedures

Joeri B. Reinders[1], Samuel E. Munoz[1,2]

[1] Leiden University College, Leiden University, The Hague, The Netherlands.

[2] Department of Civil and Environmental Engineering, Northeastern University, Boston, MA, USA.

*Correspondence to:* Joeri B. Reinders (j.b.reinders@luc.leidenuniv.nl)

**Abstract.** Flood hazard is typically evaluated by computing extreme flood probabilities from a flood frequency
distribution following nationally defined procedures in which observed peak flow series are fit to a parametric
probability distribution. These procedures, also known as flood frequency analysis, typically recommend only one
probability distribution family for all watersheds within a country or region. However, extreme flood probability
estimates ($>$ 500-year-flood or $Q_{500}$) can be biased when fit to an inappropriate distribution model because of
differences in the tails between distribution families. Here, we demonstrate that hydroclimatic parameters can aid the
selection of a parametric flood frequency distribution. We use L-moment diagrams to visually show the fit of gaged
annual maxima series across the United States, grouped by their Köppen climate classification and the precipitation
intensities of the basin, to a General Extreme Value (GEV), Log Normal 3 (LN3) and (log-)Pearson 3 (P3) distribution.
Our results show that basic hydroclimatic properties of a basin exert a significant influence on the statistical
distribution of the annual maxima. The best-fitted family distribution shifts from an GEV towards an LN3 distribution
across a gradient from colder and wetter climates (Köppen group D, continental climates) towards more arid climates
(Köppen group B, dry climates). Due to the diversity of hydrologic processes and flood generating mechanisms among
watersheds within large countries like the United States, we recommend that the selection of distribution model be
guided by the hydroclimatic properties of the basin rather than relying on a single national distribution model.





## 1. Introduction

Rivers are economic arteries that provide important resources for communities around the world, yet riverine flooding presents a major hazard for people and infrastructure along interior waterways (Mallakpour and Villarini, 2015; Peterson et al., 2013). Infrastructure, flood mitigation, and emergency response planning depend on reliable methods to compute extreme flood probabilities that typically use the probability distribution of annual maxima discharge, also known as the flood frequency distribution, from which one can compute extreme flood probabilities (e.g., the 100-year flood ($Q_{100}$), a flood with a 1% chance of occurrence in a given year) (Hamed and Ramachandro Rao, 2000; Kidson and Richards, 2005; Cassalho et al., 2019). To construct flood frequency distributions, national flood frequency procedures around the world fit annual gaged stream maxima to a parametric probability distribution model (Castellarin et al., 2012; Madsen et al., 2014). These approaches involve, and are affected by, the *a priori* assumption of what parametric statistical model best captures the empirical distribution of flood magnitudes (Kidson and Richards, 2005). Standard national procedures often prescribe only one probabilistic distribution model, for example, in the United States Bulletin 17C recommends the Log-Pearson III (LP3) distribution family (England et al., 2019), while in the United Kingdom the FEH (Flood Estimation Handbook) recommends the use of the General Logistic distribution (GLO) (Robson and Reed, 1999). Although these recommendations provide a consistent framework for flood frequency analyses (Barth et al., 2019), they can also result in biases for infrequent flood probability estimates when applied over large hydro-climatically diverse regions (Klemes, 1993; Merz and Blöschl, 2003, 2008) due to the inherent differences in the distribution shape and tail thickness of different parametric models. These differences can lead to large uncertainties for extreme flood probability estimates such as the thousand- or ten-thousand-year floods.

Watershed morphology, land use, and climatology affect the flood frequency distribution mean, variance, and tails or respectively the location, scale, and shape parameters of the probabilistic distribution. For example, large watersheds have the capacity to absorb heavy precipitation events better than small watersheds (Iacobellis et al., 2002; Salinas et al., 2014b), meaning that peak flow in smaller watersheds is disproportionally affected by an extreme precipitation event and that they observe higher peak flow variances (Iacobellis et al., 2002; Salinas et al., 2014a). Other studies have noted more complex relationships, where the variance of flood frequency distributions decreases with watershed area for small watersheds, but increases with area for large watersheds (Blöschl and Sivapalan, 1997; Smith, 1992). Urbanization leads to a reduction in soil permeability and an increase in precipitation induced surface runoff (Hall et al., 2014; Hodgkins et al., 2019), which results in more local flash floods that are associated with thicker tailed flood frequency distributions (Merz and Blöschl, 2003; Zhang et al., 2018). Relatedly, population growth (a proxy for urbanization) and river engineering (e.g., channel straightening) can increase mean annual peak flows (Villarini et al., 2009; Munoz et al., 2018).

Local flood generating mechanisms, particularly the type, duration, and intensity of local precipitation events, affect all aspects of the flood frequency distribution (Hall et al., 2014; Merz and Blöschl, 2003). In watersheds where precipitation occurs predominantly as rain as opposed to snow, flood frequency distributions exhibit higher variance (Merz and Bloschl, 2003; Gaál et al., 2015). Similarly, watersheds where total annual precipitation only falls in a few intense events also have flood distributions with high variances (Blöschl and Sivapalan, 1997; Pitlick, 1994), whereas watersheds with high total annual precipitation observe flood distributions with lower variances (Salinas et al., 2014b). Merz and Blöschl (2003) summarized several of these findings in their typology of regional flood generating mechanisms. Antecedent soil moisture adds another level of complexity to the relationship between precipitation and flood frequency distribution shape, as synchronicity between precipitation and antecedent soil moisture levels is likely to thicken the flood frequency distribution tails precipitation (Ivancic and Shaw, 2015).

These findings raise the question to what extent we can improve the accuracy of extreme flood probability estimates by selecting a distribution family that more closely reflects the underlying distribution of annual maxima. One method is to select a parametric distribution based on the value of an environmental parameter at the gaged location, for example a precipitation statistics or drainage area. Salinas et al. (2014b) demonstrated that European rivers with different drainage areas and total annual precipitation fit differently to multiple three-and-two parameter distribution families. However, as described above, the relation between drainage area and flood frequency shape is complex and annual maximum rainfall does not describe the type of precipitation. There are relatively few studies that relate flood frequency distributions to aggregated climate classifications such as the Köppen climate regions (Kottek et al., 2006; Peel et al., 2007). In one such study, Metzger et al. (2020) demonstrate that flood frequency distributions in arid and semi-arid regions give larger 10 to 100-year flood ratios compared to Mediterranean climates; a similar relation was found when arid regions are compared to humid regions (Zaman et al., 2012). These findings provide strong support for the hypothesis that the hydroclimatic properties of a basin – particularly aggregate hydroclimatic classifications





like the Köppen system – influence the tail thickness of flood frequency distributions, and thus exert considerable influence on the probabilities of extreme flood events.

Here we build on the previous work by Salinas et al. (2014a, 2014b) by examining the fit of annual maxima streamflow data from across the United States to several three-parameter distributions via L-moment diagrams. We perform a similar experiment, but group annual maxima gage records based on two aggregated hydroclimatic variables instead of one-dimensional variables: (1) the Köppen climate region, which are based on temperature and precipitation normals (Kottek et al., 2006); and (2) by watershed precipitation intensity, which is a combination of the maximum daily precipitation and the total annual precipitation. By grouping gage discharge records based on the hydroclimatic properties of their basin, we assess whether these variables can guide *a priori* parametric distribution model selection. Our results demonstrate that peak flow records from different Köppen climate regions and precipitation intensity groups tend to fit to specific distribution families. These findings imply that basic hydroclimate properties of a watershed could be used to guide the selection of a distribution family in flood frequency analysis.

## 2. Hydroclimatic Data & Methodology for the L-moment Diagrams

### 2.1 Data

We chose the United States for our study because it spans all of the five main Köppen climate groups (Peel et al., 2007) and has watersheds that are influenced by a diverse set of synoptic weather systems (Hirschboeck, 1988). Yet, despite this hydroclimatic diversity, the LP3 distribution is recommended for all flood frequency analyses in the United States in Bulletin 17C (England et al., 2019). To determine the flood frequency distribution shape of different United States rivers, we constructed a dataset containing 1538 annual maxima discharge records (Fig. 1a). This dataset is a selection of the larger USGS surface water database which contains observational data from a network of gages across the United States (USGS, 2020). To generate our dataset, we first selected all records longer than 30 years. Next, we picked the longest continuous record 7 for each independent USGS hydrologic unit, to avoid biasing the distribution selection towards more heavily sampled rivers. We also included records from Alaska and Hawaii to encompass additional hydroclimatic diversity. The annual maxima records in the final dataset have an average length of 78 years, and a range from 30 to 118 years (Fig. 1b).

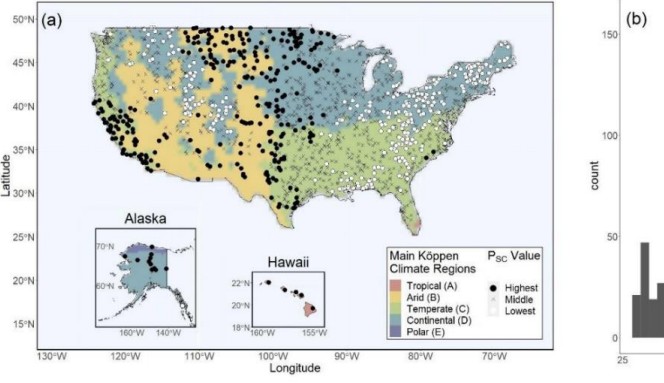
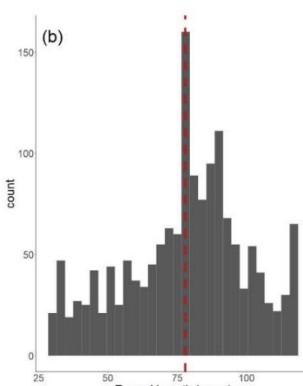

**Figure 1: (a) Locations of hydrologic annual maxima records from instrumental river gages used in this study grouped by the percentage of the maximum annual daily precipitation level by the total annual precipitation level ($P_{SC}$) and plotted atop their Köppen climate region. (b) Histogram of discharge record length in years, the red line indicates the mean value 78 years.**

To classify the gage records in different hydroclimatic groups, each annual maxima record was assigned a Köppen climate classification (Peel et al., 2007; ORNL DAAC, 2017), and a long-term (1981-2010) daily mean precipitation record from the Climate Prediction Center (CPC) precipitation dataset based on proximity (CPC, 2020). First, annual maxima records were categorized by their main Köppen climate group: arid (B), temperate (C), or continental (D) —


other climate groups did not have enough representation among the gages compared to the other climate groups (six for tropical and one for polar – all located in Hawaii or Alaska) (ORNL DAAC, 2017). Of the 1538 annual maxima, 215 are in an arid climate (Köppen group B), 578 are located in temperate climates (Köppen group C), and 738 are located in continental climates (Köppen group D). Next, we categorized annual maximum records by their watershed's hydroclimatic intensity, defined here as the percentage contribution of the maximum daily precipitation level to the

total annual precipitation ($P_{SC}$) in the CPC record (CPC, 2020). Gages close to high $P_{SC}$ values thus experience most precipitation during high intensity events, whereas gages with low $P_{SC}$ values experience precipitation more evenly throughout the year. We also assessed other precipitation metrics (e.g. annual maximum daily precipitation and the 95th percentile for daily precipitation distribution) but these metrics were not as meaningfully associated with flood distributions as $P_{SC}$, which is similar to precipitation metrics known to influence flood frequency distribution shape (Metzger et al., 2020; Pitlick, 1994). Each annual maxima record was assigned to one of three groups: the lowest 20%

$P_{SC}$ values containing 308 records (i.e., precipitation spread more evenly throughout the year), the highest 20% $P_{SC}$ values containing 308 records (i.e., a significant proportion of annual precipitation falls in one storm), and all intermediate values encompassing the remaining 922 records. The 20th and 80th percentiles were chosen, because they preserve a meaningful difference between the two groups while maintaining large sample sizes.

### 2.2 L-Moment Diagrams

We use L-moment diagrams to measure the fit of annual maxima records to parametric distributions. L-moment diagrams are a graphical tool to assess the goodness-of-fit of multiple annual maxima records to a series of probabilistic models and guide the selection of a regional flood frequency distribution family (Peel et al., 2001; Vogel and Fennessey, 1993). The L-moments of a hydrologic record are the linear combinations of its order statistics and, like regular moments (i.e. the mean, standard deviation, and skewness) describe the shape of a sample distribution. A

detailed description of how to compute L-moments is given by Hosking and Wallis (1997). L-moments are often preferred over regular moments because they are more robust to small sample sizes (Hosking, 1990; Wang, 1990). An L-moment diagram is constructed by plotting the L-moment coefficient of skewness (L-Cs) against the corresponding L-moment coefficient for kurtosis (L-Ck) (Hosking and Wallis, 1997). Consequently, any 3-parameter distribution can be plotted as a line in the L-moment diagram from their mathematical formulation of the ratio between L-Cs and

L-Ck (Table 1). The distance between the L-Cs and L-Ck of a sample and the line describing a particular 3-parameter distribution 8 represents the likelihood of the record deriving from that distribution - the closer the sample to the line, the better the fit (Hosking and Wallis, 1997).

**Table 1: Overview of the distributions used in this study including their probability density function and L-moments (ratios)**

| Name | Probability Density Function | L-moments and L-moment ratios |
|---|---|---|
| Generalized Extreme Value Distribution (GEV) | $f(x) = \frac{1}{\alpha} e^{-(1-k)y - e^{-y}}$, $y = \begin{cases} -\frac{1}{k}\log\left\{1 - k\left(\frac{x-\xi}{\alpha}\right)\right\}, k \neq 0 \\ \frac{x-\xi}{\alpha}, k = 0 \end{cases}$ | $\lambda_1 = \xi + \alpha\frac{\{1 - \Gamma(1+k)\}}{k}$, $\lambda_2 = \frac{\alpha(1 - \frac{k}{2})\Gamma(1+k)}{k}$ $\tau_3 = 2\frac{(1 - 3^{-k})}{(1 - 2^{-k})} - 3$ $\tau_4 = \frac{5(1 - 4^{-k}) - 10(1 - 3^{-k}) + 6(1 - 2^{-k})}{(1 - 2^{-k})}$ |
| Log-Normal 3-parameter Distribution (LN3) | $f(x) = \frac{e^{ky - y^2/2}}{\alpha\sqrt{2\pi}}$, $y = \begin{cases} -\frac{1}{k}\log\left\{1 - k\left(\frac{x-\xi}{\alpha}\right)\right\}, k \neq 0 \\ \frac{x-\xi}{\alpha}, k = 0 \end{cases}$ | $\lambda_1 = \xi + \alpha\frac{\{1 - e^{\frac{k^2}{2}}\}}{k}$, $\lambda_2 = \frac{\alpha}{k} e^{\frac{k^2}{2}}\left\{1 - 2\Phi\left(\frac{-k}{\sqrt{2}}\right)\right\}$ |
| Pearson Type III Distribution | $f(x) = \frac{(x-\xi)^{\alpha-1} e^{-(x-\xi)/\beta}}{\beta^\alpha \Gamma(\alpha)}$ $\alpha = \frac{4}{\gamma^2}$ $\beta = \frac{1}{2}\sigma|\gamma|$, $\gamma \neq 0$ $\xi = \frac{\mu - 2\sigma}{\gamma}$ | $\lambda_1 = \xi + \alpha\beta$, $\lambda_2 = \pi^{-\frac{1}{2}}\beta\frac{\Gamma(\alpha + \frac{1}{2})}{\Gamma(\alpha)}$ $\tau_3 = 6I_{\frac{1}{3}}(\alpha, 2\alpha) - 3$ |

Notes: Formulation of the probability density functions, and L-moments are taken from Hosking and Wallis (1997). For all functions ξ, α, and k are the location, scale and shape parameters, respectively. When τ3, τ4 are determined via 465 a rational function approximated, which is described in detail in Hosking and Wallis (1997).






The L-moments and L-moment ratios for each annual maxima record in our dataset are compared to a GEV, LN3 and a Pearson 3 (P3) distribution. These three parameter distributions are commonly used in hydrologic sciences (Salinas et al., 2014b), and are known to fit extreme flood values in the United States well (Vogel et al., 1993; Vogel and Wilson, 1996). Additionally, we plotted Log-transformed discharge records in L-moment diagram, to fit them to a
LP3 distribution. L-moment diagrams are constructed for all records in the dataset, and each selection of record based on their Köppen climate classification and $P_{SC}$ value.

Prior work demonstrated that selecting one distribution that provides the best fit to annual maxima is difficult over a large hydrologically heterogeneous region due to the high sample variance of the L-moments (Asikoglu, 2018; Salinas et al., 2014a). To reduce the noise and guide model selection, we compute a weighted moving average of neighboring
L-Cs and their corresponding L-Ck proportional to record length. Salinas et al. (2014a) applied this method to annual maxima series from across Europe to argue for the GEV distribution as a pan-European flood frequency distribution. We computed the weighted moving average and its 95% confidence interval to summarize sample variance and facilitate distribution selection of all L-moment diagrams in this study. The weighted averages are taken from 50 consecutive L-Cs's and of the 50 corresponding L-Ck's proportional to record length.

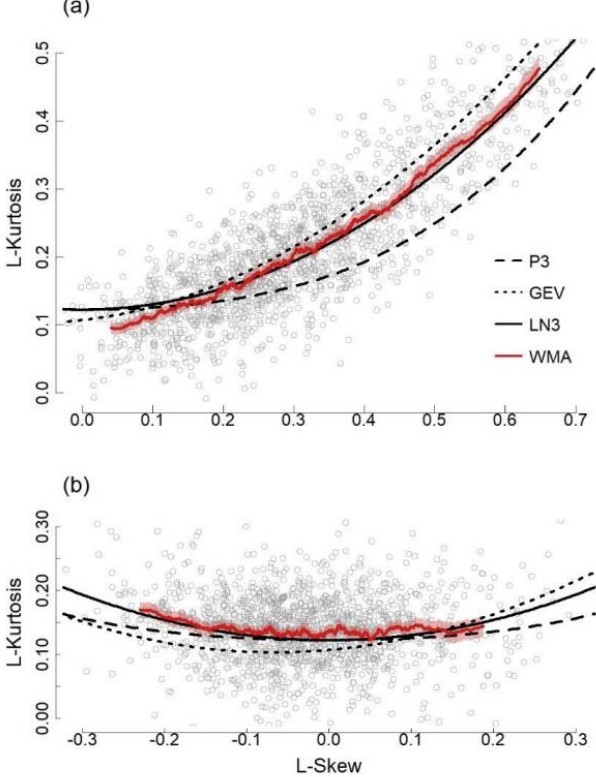


**Figure 2: L-moment diagrams with the L-moment ratio for skew and kurtosis of annual maxima records used in this study (gray dots; n=1538), with their weighted moving average (WMA) proportional to record length (red line) and the P3, GEV and LN distribution lines: (a) annual discharge maxima as recorded by the gage, and; (b) the logarithm of the annual discharge maxima.**




## 3. Results

The weighted moving averages demonstrate that the average statistical properties of the 1538 L-moment ratios across the United States are best characterized by the LN3 distribution, with large variance among individual records (Fig. 2). Specifically, the weighted moving averages of the largest L-Cs and L-Ck follow the LN3 distribution line as

opposed to the P3 and GEV distributions (Fig. 2a). Generally, these originate from rivers for which the discharge of the extreme flood is relatively large compared to the average annual flood peak. The weighted moving average deviates from the LN3 distribution for smaller L-moment ratios, which are better characterized by the GEV distribution (Fig. 2a). The theoretical distribution lines are more clustered for smaller L-moments, reflecting the similarities of GEV and LN3 distributions for values centered around the mean (Fig. 2a). In log-space, clustering of

L-moment ratios around the LP3 distribution supports the general use of this distribution for rivers in the United States (England et al., 2019), although we note that the data more closely track the LN3 distribution, particularly for extreme values of skew and kurtosis (Fig. 2b).

When the annual maxima are grouped by their respective Köppen climate region, the weighted moving average shifts to track different distribution families more closely (Fig. 3). The statistical properties of records from arid climates

are best described by an LN3 or in some cases, even the P3 distribution (Fig. 3a), whereas those from continental regions are represented by a GEV distribution (Fig. 3e) – though we note that the smaller sample size of the arid climate group results in larger confidence intervals. The weighted moving average of annual maxima records from temperate climates follows the LN3 distribution (Fig. 3c). The concentrations of individual L-moment ratios also shift when grouped by climate group: the clustering of L-moment ratios for continental climates is highest along the GEV

distribution line (Fig. 3e), for temperate L-moment ratios it falls in between the GEV and LN3 line (Fig. 3c), and for arid L-moments between the LN3 and P3 line (Fig. 3a). This shift is not observed in log-space, as log-transformed records in arid climates exhibit relatively small L-Ck compared to records from continental and temperate climates and are thus better represented by the GEV distribution (Fig. 3b). Continental and temperate regions are both well represented by the LP3 and LN3 distribution (Figs. 3d and 3f).

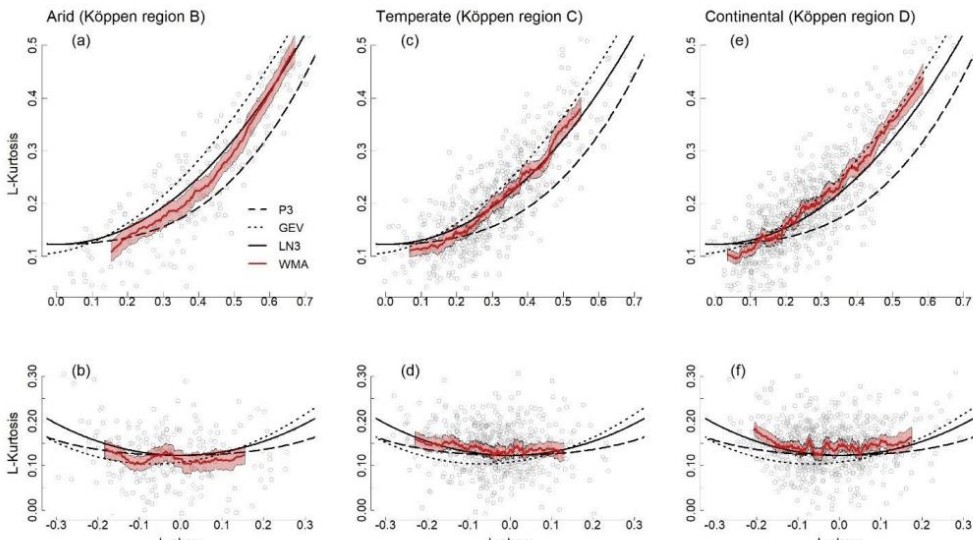


**Figure 3: L-moment diagram with the L-skew and L-kurtosis for annual discharge maxima records (gray dots) grouped by their Köppen climate region, their weighted moving averages (WMA) proportional by record length (red line) and the P3, GEV and LN distribution line (striped; dotted; solid). Panels (a), (c), and (e) show annual discharge maxima as recorded by the gage; Panels (b), (d), and (f) show the logarithm of the annual discharge maxima.**




The categorization of discharge records based on their precipitation intensity $P_{SC}$ value also result in differences between the weighted moving averages (Fig. 4). The GEV distribution most closely reflects the statistical properties of flood frequencies from watersheds with low $P_{SC}$ values (Fig. 4a), whereas high values follow the LN3 distribution line (Fig. 4e). The weighted moving average of the intermediate group falls in between the GEV and LN3 distribution lines, indicating a shift from the GEV to the LN3 distribution as $P_{SC}$ values become higher (Fig. 4b). Additionally, the range of L-Skew values is much smaller for records with low $P_{SC}$ values (Fig. 4a). We found no strong contrast between the groups when records are log-transformed (Figs. 4b, 4d, and 4f); the weighted moving averages of all three groups follow the LN3 distribution line.



**Figure 4: L-moment diagram with the L-skew and L-kurtosis for gage discharge records (gray dots) grouped by their percentage of the maximum annual daily precipitation level by the total annual precipitation level ($P_{SC}$), their weighted moving averages (WMA) proportional by record length (red line) and the P3, GEV and LN distribution line. Panels (a), (c), and (e) show annual discharge maxima as recorded by the gage; panels (b), (d), and (f) show the logarithm of the discharges.**

Our analyses document a pattern for both the Köppen climate groups and the $P_{SC}$ groups, where arid climates (high $P_{SC}$) and continental climates (low $P_{SC}$) move away from the LN3 distribution towards the P3 and GEV distribution. In contrast, the weighted moving averages of both the temperate and intermediate $P_{SC}$ group trend closer to the LN3 distribution line. A major difference between these two categories is the range of L-skew values between the corresponding groups. For example, the range of the lower $P_{SC}$ is smaller than that of the L-moment ratios in continental climates. In general, our results point to the LN3 distribution as the best distribution to characterize annual maxima data from across the United States, but also demonstrate how regional hydroclimatic differences explain part of the variance among individual flood distributions.

## 4. Discussion

The main objective of this study is to evaluate whether hydroclimatic information can improve extreme flood probability estimates in flood frequency analysis procedures. To do this, we grouped annual hydrologic maxima from gage records across the United States by their hydroclimatic properties, and used L-moments to guide the selection of a probability model. Our work provides insights into the hydroclimatic parameters that drive flood frequency distribution shape, and how to supplement conventional flood frequency analysis using hydrological information accordingly.





### 4.1 Flood Frequency Distributions in the United States

Our analyses illustrate that different Köppen climate regions (Fig. 3) and precipitation intensities (Fig. 4), generate shifts in the annual maxima L-moment samples to track different parent distributions. These findings contribute to
prior work by Salinas et al., 2014a, 2014b), who constructed an L-moment diagram for European gage records and argued for the GEV as a pan-European distribution. Salinas et al. (2014b) demonstrated with a simulation experiment, that the high variance of the L-moment ratio samples cannot alone be attributed to sampling error, and that other covariates were needed to explain the variance. While our work does not prove a causal relation between Köppen climate region, precipitation intensity, and flood frequency distribution shape, we do demonstrate that hydroclimatic
factors explain part of the L-moment sample variance and flood frequency distribution shapes across the United States. In our study, we demonstrate that the average statistical properties of annual hydrologic maxima across the United States are most closely represented by the LN3 distribution — although sample variance remains high (Fig. 2). These findings are consistent with, but further specifies, prior work by Vogel and Wilson (1996), who used L-moment diagrams to conclude that the LN3, LP3, and GEV distributions are all reasonable representations of annual maxima
across the United States.

Our findings demonstrate that the distribution family that best characterized hydrologic maxima shifts from the GEV towards the LN3 distribution as we move from colder and wetter climates (Köppen group D) to arid and drier climates (Köppen group B) (Fig. 3). The contribution of the annual maximum storm to annual total precipitation ($P_{SC}$) also
reflects this pattern: watersheds with a lower maximum storm contribution are best captured by the GEV distribution and with a higher $P_{SC}$ by the LN3 distribution (Fig. 4). These findings are consistent with the shift of the best-fit distribution from GEV to LN3 as the total annual precipitation decreases over a catchment observed by Salinas et al. (2014b) in Europe. Our results also support findings of Pitlick (1994) who showed that local flood frequency distributions in mountainous areas of the Western United States are shaped by regional precipitation intensity. In our
study, annual maxima in arid regions are best described by the LN3 distribution where there are large differences in the ratio of mean to extreme precipitation rates (Metzger et al., 2020; Zaman et al., 2012), whereas temperate regions, with precipitation spread out more evenly throughout the year, are best fit by the GEV distribution. Köppen climate regions also provide a potential explanation for why Salinas et al. (2014b) found the GEV distribution to be the best fit for European annual maxima, as it is a continent dominated by temperate and cold climates (Köppen groups C and
D). In contrast, the United States includes large regions with arid climates as well as temperate and cold regions, which shifts the overall best fit distribution for annual maxima to the LN3 distribution.

### 4.2 Improvements to Flood Frequency Analysis

Our findings point to climate classification schemes that encompass multiple variables, for instance the Köppen climate classification, as an effective means for regional probability model selection in flood frequency analysis. Other
factors that influence flood frequency distributions — including precipitation seasonality and intensity, vegetation, soil type, infiltration capacity, and surface runoff levels —are indirectly included within the Koppen classification scheme, as it is based on temperature and precipitation levels, which in turn affect these variables (Kottek et al., 2006; Peel et al., 2007). Such, the observed results for gages grouped by Köppen climate regions are likely confounded by any of these factors, including $P_{SC}$. Climate classification schemes provide another benefit as they encompass large
contiguous areas. Local precipitation intensities may represent large river systems poorly if discharge at given downstream location could be influenced by multiple precipitation regimes from multiple tributaries. Encountering this problem becomes less likely with Köppen regions which often cover entire watersheds. However, as with any generalization, regional context should not be overlooked as hydroclimate can still be diverse even within Köppen regions (Kottek et al., 2006; Peel et al., 2007).

Our main finding – that hydroclimatic properties of a basin exert a strong influence on the distribution of annual discharge maxima – provides a simple means to improves the accuracy of extreme flood probability estimates without altering the mathematical procedure described in flood frequency analysis guidelines like Bulletin 17C (England et al., 2019). One approach to further improve on our work is the weighted mixed populations framework, where one
stratifies data and fits a parametric distribution to each new data population to aggregate the population distributions into a single distribution weighted on population size (Barth et al., 2019). In a hydrologic context, one could subdivide annual flood discharges based on different (periodic) flood-generating mechanisms. Accordingly, this method works particularly well for watersheds with multiple distinct flood-generating mechanisms, for example due to periodic atmospheric rivers, and skewed flood distributions (Barth et al., 2019). Another approach is to use other parametric
distributions, with four or more parameters – although such methods do not explicitly consider hydrologic information – or a Metastatistical Extreme Value Distribution (MEVD) (Marani and Ignaccolo, 2015; Miniussi et al., 2020). A



MEV distribution derives an extreme (annual maxima) flood frequency distribution via 'ordinary' discharge values and has shown to be efficient with all sorts of parametric distributions (Marani and Ignaccolo, 2015). Finally, our results do not preclude the utility of other basin characteristics, specifically geomorphic properties of the watershed

(e.g., size and elevation) in further improving probability model selection, although prior work points to hydroclimatic variables as exerting a primary control on flood distributions.

## 5. Conclusions

We evaluated annual hydrologic maxima distributions from across the United States, and showed that probability model selection can be improved when it is based on the hydroclimatic properties of the basin. In the United States, the weighted moving average of L-moment ratios follows the LN3 distribution, and this distribution could serve as national distribution family. However, distribution selection can be improved by taking a basin's climate region into account, where continental climates (cool/wet) are best described by GEV distributions while arid climates (hot/dry)

are best described by LN3 distributions. More broadly, our work demonstrates that the climatology of a region is a powerful tool to guide the a priori distribution selection in flood frequency analysis.

### Acknowledgments
We would like to thank Willem Toonen, Paul Hudson, Ed Beighley, Auroop Ganguly, Dick Bailey for valuable

discussion and comments on this work. This project was supported by grants from the US National Science Foundation (EAR-1804107 and EAR-1833200).

### Code availability
The R-scripts used to perform the analyses and make the figures in this paper are available from the Zenodo open

respiratory https://doi.org/10.5281/zenodo.4476066. 12

### Data availability
Data in this study are available through the United States Geological Survey (USGS) Water Data for the Nation (https://waterdata.usgs.gov/nwis). The specific dataset containing the 1538 USGS peak flow records is available via

the Zenodo open respiratory https://doi.org/10.5281/zenodo.4476066.



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

**Table 1: Overview of the distributions used in this study including their probability density function and L-moments (ratios)**

| Name | Probability Density Function | L-moments and L-moment ratios |
|------|------------------------------|-------------------------------|





| Distribution | PDF | L-moments |
|---|---|---|
| **Generalized Extreme Value Distribution (GEV)** | $f(x) = \frac{1}{\alpha} e^{-(1-k)y - e^{-y}}, y = \begin{cases} -\frac{1}{k} log\left\{1 - k\left(\frac{x-\xi}{\alpha}\right)\right\}, k \neq 0 \\ \frac{x-\xi}{\alpha}, k = 0 \end{cases}$ | $\lambda_1 = \xi + \alpha \frac{(1 - \Gamma(1+k))}{k}, \quad \lambda_2 = \frac{\alpha\left(1 - \frac{k}{2}\right)\Gamma(1+k)}{k}$ <br> $\tau_3 = 2\frac{(1-3^{-k})}{(1-2^{-k})} - 3$ <br> $\tau_4 = \frac{5(1-4^{-k}) - 10(1-3^{-k}) + 6(1-2^{-k})}{(1-2^{-k})}$ |
| **Log-Normal 3-parameter Distribution (LN3)** | $f(x) = \frac{e^{ky - y^2/2}}{\alpha\sqrt{2\pi}}, y = \begin{cases} -\frac{1}{k} log\left\{1 - k\left(\frac{x-\xi}{\alpha}\right)\right\}, k \neq 0 \\ \frac{x-\xi}{\alpha}, k = 0 \end{cases}$ | $\lambda_1 = \xi + \alpha \frac{\left(1 - e^{\frac{k^2}{2}}\right)}{k},$ <br> $\lambda_2 = \frac{\alpha}{k} e^{\frac{k^2}{2}} \left\{1 - 2\Phi\left(\frac{-k}{\sqrt{2}}\right)\right\}$ |
| **Pearson Type III Distribution** | $f(x) = \frac{(x-\xi)^{\alpha-1} e^{-(x-\xi)/\beta}}{\beta^\alpha \Gamma(\alpha)}$ <br> $\alpha = \frac{4}{\gamma^2}$ <br> $\beta = \frac{1}{2}\sigma|\gamma|, \gamma \neq 0$ <br> $\xi = \frac{\mu - 2\sigma}{\gamma}$ | $\lambda_1 = \xi + \alpha\beta, \quad \lambda_2 = \pi^{-\frac{1}{2}}\beta \frac{\Gamma(\alpha + \frac{1}{2})}{\Gamma(\alpha)}$ <br> $\tau_3 = 6I_{\frac{1}{3}}(\alpha, 2\alpha) - 3$ |

Notes: Formulation of the probability density functions, and L-moments are taken from Hosking and Wallis (1997). For all functions ξ, α, and k are the location, scale and shape parameters, respectively. When τ3, τ4 are determined via 465 a rational function approximated, which is described in detail in Hosking and Wallis (1997).
