# Peer review of "Accounting for Hydroclimatic Properties in Flood Frequency Analysis Procedures"

_Hydrology and Earth System Sciences, 2022_

## Author Comment (AC1)

**Author Comment: Accounting for Hydroclimatic Properties in Flood Frequency Analysis Procedures**

*Joeri B. Reinders and Samuel E. Munoz*

**Reviewer 1**

I totally agree with the importance of the objective of this paper trying to shed some light on the distribution selection for flood frequency analysis, and with the main conclusion that "probability model selection can be improved when it is based on the hydroclimatic properties of the basin". It is a concise paper and I have enjoyed reading it. Some questions from my side:

*We thank Reviewer 1 for the supportive comments.*

In Fig 2 and its derivations in terms of Kopen index and Psc, the WMA clearly helps to see the potential similarities with the theoretical L-moments of the 3 distributions, but what do the WMA confidence limits really provide?

In all figures (or in one representative) I am missing some "better uncertainty", such as that of the sample L-moments, in order to explain their dispersion. However, they are a function of the sample size, so it will be difficult to deal with it. What is your opinion?

*We thank Reviewer 1 for these comments on the uncertainty measure in our work and combined the two points in one reply. The limits provide the 95%CI for the WMA – so not an uncertainty that relates to the spread of the sample L-moments – but rather the spread of the L-Ck's per bin in the WMA analysis. A better indicator of uncertain over the sample L-moments itself could be provided by adding the standard deviation over sections of L-sk. This would also be in line with helpful comments made by Reviewer 2 who suggested taking into account the variation along the skew axis.*

In section 4.2 authors define the following steps in their research, but it would not be very complex to advance part of them. I particular

- Why only these 3 distributions with 3 parameters? Excluding the 4-parameter ones and mixed distributions (I would mention also the TCEV), I am curious about the potentiality of the 3-parameter Generalized Pareto.

*We think these would be very interesting additions and can add these in an updated manuscript.*

- You have used as explanatory variables the Kopen and the Psc and I totally agree with the conclusions in section 3. Did you try the catchment area, as mentioned its importance in L50 and 69?

*We did try catchment area in a preliminary study however this did not give results that were significantly different from those in figure 2. However, we can add these in an appendix.*

Very minor comments:

L17. (log-)Pearson 3 (P3) or Pearson 3 (P3)? I.e., (log-) is confusing.

*Thank you for pointing this out, we will adjust this accordingly.*

L25 and 26 are not needed. Too often authors (myself included) try to give a general framework that is too general and unnecessary.

*Thank you for pointing this out, we will adjust this accordingly.*

***Reviewer 2***

Thank you for the opportunity to review the subject-manuscript. In this study, the authors evaluate the effects of accounting for hydroclimatic properties in flood frequency procedures. While the authors evaluate the importance hydroclimatic information for various flood frequency distributions in the hydrologically diverse United States, the general approach of climate regions may be too broad, resulting in an overgeneralized description of the best candidate distributions across the U.S. Additional comments are also provided to improve the overall study and findings.

*We thank Reviewer 2 for the extensive comments on our manuscript. We agree that climate regions (and precipitation) are broad categories for flood distributions. However, in practice procedures for flood frequency analysis often apply only a single parametric distribution – and this work is to show that generalities like climate and precipitation can be used to provide higher accuracy. We do agree with many of the comments from Reviewer 2 about a more nuanced description of our results, especially with regard to trend in the skew. We will adopt these changes to better represent the debate on the usability of broad categories within flood frequency analyses.*

Specific/technical comments:

In the abstract and Introduction sections, why do the authors describe evaluating the performance of 500-year flood events? Does this seem reasonable given the more common lengths of annual peak discharge records that are commonly less than 75 years of record? Furthermore, why do the authors state that we can estimate a 1,000 or 10,000-year flood event based on these records and draw conclusions about candidate distributions this far in the tails when we do not know the true distribution? These descriptions seem to counter what the author's used for their analysis: "The annual maxima records in the final dataset have an average length of 78 years and a range from 30 to 118 years." Therefore, recommend revising the manuscript to describe drawing inference for the performance of the candidate distributions among the heavy tails closer to the 100-[at most] the 500-year annual exceedance probabilities.

*We agree with the important point that Reviewer 2 makes about the usability of relatively short observational records for the analysis of extreme flood probabilities. We agree that the records used in this analysis would be insufficient to accurately estimate for example 1,000 or 10,000-year floods. Yet, frequency analyses are used to compute flood probabilities of these large return periods – whether it be for infrastructure design purposes or to estimate the return period of an extreme event. We agree the abstract could be rewritten to better reflect the large uncertainties associated with these type of analyses – something we try to address with this research.*

As shown in Figure 2b, for skews ~ -0.1 to 0.2, the LP3 performs well just as the LN3. However, the LP3 performs poorly for negative skews <-0.1. This is why B17C proposes the LP3 distribution, but with the expected moments algorithm (EMA) to estimate the moments. It has long been recognized that there are statistically low outliers in a block maxima approach (e.g., selecting one annual peak per year) for a flood frequency analysis. To handle low outliers, B17C, recommends using the multiple Grubbs-Beck test (MGBT) to detect these statistical low outliers (e.g., potentially influence low floods (PILFs)) that may have undue influence on the upper right-hand tail (e.g., low exceedance probabilities). If PILFs are detected, those flows are recoded as censored flows and the EMA method is used to estimate the parameters of the LP3 distribution. If a more robust statistical test for the identification and treatment of PILFs is not employed, the LP3 distribution will preform poorly, especially for those very negatively skewed distributions which are found across CONUS and not just in the arid southwest U.S., for example. Given this major caveat when using the LP3 distribution, recommend this study take that into account when describing the results of this study. And recognize

the current approach for the LP3 in this study does not honor the updates to the Federal guidelines to better fit the agreed-upon use of the LP3 in the U.S.

*We agree with Reviewer 2 – this is a valuable point that should be added to the discussion on the use of the LP3 distribution.*

Also, because we do not know the true distribution of the annual maxima series, these caveats need to be address and discussed.

Do not think there are any consistent better distributions throughout the range of skews in the arid region. Even in the log transformed space there are no clear results of better fits. This is most likely attributed to the additional diversity among the arid region. There are regional differences between central and southern CA, the southern four corners region vs. the Midcontinent region west of the 100th meridian. Those two regions are not commonly grouped together.

*We agree with Reviewer 2 that accounting for the diversity within climate zones would result in more accurate fitted distributions. However, the purpose of this paper was to explore whether broad divisions like climate and precipitation characteristics would result in specific conclusions regarding the use of one distribution over another. We agree however that the results could be adjusted to account for the nuance made by Reviewer 2.*

*We could run a similar analysis, for example, based on the regions used in the National Climate Assessment, which would better account for some of the diversity described by Reviewer 2.*

Disagree with this description: "This shift is not observed in log-space, as log-transformed records in arid climates exhibit relatively small L-Ck compared to records from continental and temperate climates and are thus better represented by the GEV distribution (Fig. 3b)." Again, it depends on what range of skews among the arid group positive, near zero and negative.

*We agree with Reviewer 2 for the same reasons as explained in the previous comments.*

Disagree with this description: "Continental and temperate regions are both well represented by the LP3 and LN3 distribution (Figs. 3d and 3f)." Recommend splitting this discussion. Temperate is best represented by X distribution over different negative/positive skews. And the same type of description for the continental region. If these recommended descriptions are used, likely differences will become more apparent.

*We agree with Reviewer 2 and will adjust this accordingly.*

Recommend adding more thorough descriptions about positive/negative skews and the best fit of the distributions.

*We agree with Reviewer 2 and will adjust this accordingly.*

Do not necessarily agree that the LN3 is the best distribution for the highest Psc (figure 4e). More commonly the LN3 is higher than the upper confidence bounds but is overall closer than the P3. Recommend revising the description.

*We agree with the nuance described by Reviewer 2 and will adjust this accordingly.*

Disagree with this description: "The weighted moving average of the intermediate group falls in between 205 the GEV and LN3 distribution lines, indicating a shift from the GEV to the LN3

distribution as PSC values become higher (Fig. 4b)." Should this description be for figure 4c? Also, as values increase the better fits goes from the LN3 (0.2-0.45) then to the GEV (~>0.45).

*This description describes the change seen from figure 4a to 4c. We agree with Reviewer 2 that a nuance on the skew parameter is in place.*

Disagree with this description: "We found no strong contrast between the groups when records are log-transformed (Figs. 4b, 4d, and 4f); the weighted moving averages of all three groups follow the LN3 distribution line." Only the middle Psc values exhibit this description. The WMA for the Lowest Psc are substantially higher than the three distributions (4b), especially for skew >-0.1. And while the LN2 performs better (4f), after ~ >0.1, all distributions appear higher.

*We agree with Reviewer 2 that the description of both 4b and 4f could be made more specific. It is true that 4b only follows the LN3 line for the lowest skew values. In 4f we would argue that the LN3 reflects the distributions well up till a skew of 0.5 – but we agree that it is important to make this point.*

Recommend rewording the overgeneralized descriptions for figures 3 and 4.

*We agree with Reviewer 2 and will make the adjustments accordingly.*

Disagree with this description: "In general, our results point to the LN3 distribution as the best distribution to characterize annual maxima data from across the United States, but also demonstrate how regional hydroclimatic differences explain part of the variance among individual flood distributions." Again, this is too generalized and doesn't reflect a combination of results based on different skews.

*We agree with Reviewer 2 as described in the previous comments. These changes in the results will be further translated into the discussion.*

"While our work does not prove a causal relation between Köppen climate region, precipitation intensity, and flood frequency distribution shape, we do demonstrate that hydroclimatic factors explain part of the L-moment sample variance and flood frequency distribution shapes across the United States." Recommend that the author's attempt to provide some causal mechanisms, especially because the climate regions seem too broad. Recommend at least finding supporting literature of other cluster methods that have found similar regions.

*We agree with Reviewer 2 and will make the adjustments accordingly.*

Disagree with this description: "In our study, we demonstrate that the average statistical properties of annual hydrologic maxima across the United States are most closely represented by the LN3 distribution — although sample variance remains high (Fig. 2)." Recommend the author's further address the sample variances and how they relate to skew ranges and the corresponding better performing distributions.

*We agree with Reviewer 2 and will make these changes.*

"Our findings demonstrate that the distribution family that best characterized hydrologic maxima shifts from the GEV towards the LN3 distribution as we move from colder and wetter climates (Köppen group D) to arid and drier climates (Köppen group B) (Fig. 3)." Too vague and disagree with the Arid regions overall description. In Figure 3a (the Arid climate region B), the WMA is between

LN3 and the P3 for skew values < 0.45. Again, the arid region is likely too broad and PILFs (low outliers) were not accounted for.

*As discussed above we agree with Reviewer 2 on that these results/ discussion points can be more precisely formulated to take into account differences along the skew values.*

"Our results also support findings of Pitlick (1994) who showed that local flood frequency distributions in mountainous areas of the Western United States are shaped by regional precipitation intensity." This finding has not been talked about nor have any results been discussed specifically related to the mountainous region in the western United States. Often high-elevation sites in the western U.S. have mixed populations that strongly deviate from one particular distribution (e.g., annual peaks generated from rainfall, snowmelt and rain-on-snow. These sites can have s-shaped hooks in the right-hand tail, for example. See B17C and other references for a come complete description of mixed populations in the western U.S. These high elevation sites are not broadly related to regional precipitation intensity.

*Here we did not mean to say that our results explain specifically distribution variations in mountains areas, but only that the precipitation intensity explains some of the variance in distribution shape as was also shown by Pitlick (1994) for mountainous watersheds in the Western United States. We agree with Reviewer 2 that these results can be formulated more precisely and that we should take into account the broader research on high elevation sites. We will adapt the discussion accordingly.*

Disagree with this description: "Climate classification schemes provide another benefit as they encompass large contiguous areas...Encountering this problem becomes less likely with Köppen regions which often cover entire watersheds." Large regions, such as the Arid region in this study, is likely too broad and oversimplifies various seasonal controlling factors on precipitation and flood generating attributions, for example.

*We agree with Reviewer 2 that there is a lot of variation within climate regions. The aim of this work is to identify whether these broad classifications can help to make flood frequency analysis within national procedures more adaptable. That is why we did not account for these variations, but we do agree we could describe how such an analysis could account for further detail.*

---

## Referee Report (RR1)

Reinders and Munoz, 2023, "Accounting for Hydroclimatic properties in flood frequency analysis procedures"

Summary: This study explores annual maxima discharge from gages across the U.S. and uses L-moments to aid selection of probability models for flood frequency analysis. They show that climatic regime and precipitation intensity of the region are useful indicators for guiding selection of the model, with cool climates best represented by GEV distributions and arid climates best represented by LN3 distributions. Overall, this is an interesting and useful study that provides nuance on flood frequency analysis in the U.S.

General comments:
- I agree with a previous reviewer that the climatic regions are quite large, but I think this is OK, as it is a starting point for showing that the LN3 distribution used for flood frequency analysis is not appropriate everywhere in the U.S.
- Where do the highest and lower 20% of Psc tend to occur spatially? It would be great to see of map of this. I assume the higest 20% of Psc occur in the arid areas, as you mention in the paper, but it would be interesting to see if that is truly the case or not.

Specific comments:
- Lines 325-335: I think simply including elevation could be a helpful way to delineate the arid region, as high elevation sites have very different precipitation and soil properties compared to low elevation prairies. This would be a great avenue for future work.

Technical corrections:
- Line 248: replace "does indicating" with "indicates".

---

## Referee Report (RR2)

[referee-annotated manuscript omitted]

---

## Author Response (AR3)

**Author Comment: Accounting for Hydroclimatic Properties in Flood Frequency Analysis Procedures**
*Joeri B. Reinders and Samuel E. Munoz*

*Reviewer 3*

*We thank Reviewer 3 for their supportive feedback and have incorporated their comments into the final manuscript.*

Reinders and Munoz, 2023, "Accounting for Hydroclimatic properties in flood frequency analysis procedures"

Summary: This study explores annual maxima discharge from gages across the U.S. and uses L-moments to aid selection of probability models for flood frequency analysis. They show that climatic regime and precipitation intensity of the region are useful indicators for guiding selection of the model, with cool climates best represented by GEV distributions and arid climates best represented by LN3 distributions. Overall, this is an interesting and useful study that provides nuance on flood frequency analysis in the U.S.

General comments:
• I agree with a previous reviewer that the climatic regions are quite large, but I think this is OK, as it is a starting point for showing that the LN3 distribution used for flood frequency analysis is not appropriate everywhere in the U.S.

• Where do the highest and lower 20% of Psc tend to occur spatially? It would be great to see of map of this. I assume the highest 20% of Psc occur in the arid areas, as you mention in the paper, but it would be interesting to see if that is truly the case or not.

*The spatial distribution of Psc values are shown on the map of figure 1. The highest 20% indeed occurs predominantly along drier climates in the Northern and Western US. The lowest 20% is predominantly located in the cooler Northeast – but not exclusively.*

Specific comments:
• Lines 325-335: I think simply including elevation could be a helpful way to delineate the arid region, as high elevation sites have very different precipitation and soil properties compared to low elevation prairies. This would be a great avenue for future work.

*We agree with Reviewer 3 that this would be a possible future direction of research and address this more explicitly in the discussion in line 339.*

Technical corrections:
• Line 248: replace "does indicating" with "indicates".

*We adjusted this accordingly.*

Review date: 2023-09-02.

This is an interesting and generally well-written paper on a topic continuing practical interest and need based on research that appears to be mostly well done. Another reviewer has stated that the regions are rather large and likely heterogeneous; I agree that may be a concern, but my major concerns are prior to that issue, since they relate to station selection and handling of the peak-flow data from those stations. Some of my concerns may turn out to be void when further details on how these issues were handled are added to paper, but the lack of such detail is a problem in itself.

*We begin by thanking Referee 4 for their thorough evaluation of our manuscript and their valuable recommendations for additional references. In the following sections, we will delve deeper into some of their comments. Nevertheless we have made sure to explore the suggestions made by Reviewer 4, with the resulting data being included in the supplementary materials.*

My two primary concerns are as follows:

1. Are stations selected filtered for the effects of regulation (for example by flood-control reservoirs) and urbanization? The effects of urbanization are addressed in the introduction, but then the topic disappears from the paper and no mention of filtering for such effects is made. Reservoirs, especially those designed for flood control and any reservoir will large storage capacity are well-known to have substantial effects on peak-flow distributions (see, for example, FitzHugh and Vogel, 2011), and stations with substantial reservoir effects thus should also be filtered out of the dataset. (A related concern is that such stations often have trends due to changes in urbanization over their period-of-record or because one or more reservoirs was built during the period-of-record.) The dataset at the Zenodo link under the Data availability statement includes several stations with which I am familiar that have substantial effects from urbanization and/or regulation, but that dataset has 4202 stations in it, while the authors say they used a dataset of 1538 stations, so maybe they did filtering that is not discussed. (The Data availability statement itself says the file containing the 1538 stations used is available at the Zenodo link; that is as it should be, but the statement appears to be false.)
An important practical issue in doing such filtering is how. The simplest suggestion I have is to use the GAGES-II dataset (Falcone, 2011). It includes basin characteristics and basin boundaries for more than 9000 gauging stations in the United States, including information on dams and other regulation, land development and impervious surfaces. More than 2000 of the stations are designated as being of "reference" condition, meaning having the least effects from human disturbance. However, the authors might also use the given characteristics to select stations using a somewhat different criterion.

2. How did the authors handle the various "peak streamflow qualification codes" that are associated with USGS peak-flow data? Those of concern for this research are code 1, which indicates that the discharge value is a maximum daily average (which implies the true instantaneous peak is likely higher); code 4, which indicates that the discharge is less than the indicated value which is the minimum recordable discharge at the site, code 8 which indicates that the discharge is actually greater than indicated value; code 7, which indicates that the peak is an historic peak; and code O, which indicates an "opportunistic" value not from systematic data collection. The code 7 and code O peaks can be simply removed (code O because they are non-systematic by definition and code 7 because they are typically very large peaks whose values were inferred from historic records and are thus also non-systematic); for the other codes, authors should consider the sensitivity of their results to the censoring that is indicated, and act accordingly. For example, it would bias the record to remove the code 4 peaks since they are generally the smallest peaks, but the values provided are

biased upwards; in this case, one relevant question is how sensitive the results are to biases in such small peaks.

*Here's the improved text with spelling and grammar checks:*

*First, we need to apologize for the confusion regarding the dataset uploaded to the Zenodo repository. Indeed, the file contains all available records longer than 30 years, not the 1538 stations that were used in our analysis (the longest record in each HUC). We describe how we selected these records in the text (first paragraph of the method section) and have added the 1538 record dataset to the repository.*

*We would like to address both comments 1 and 2 in one go as they relate to each other:*

*In this research, we did not include urbanization as an independent variable, as this study focuses on the use of an institutionalized distribution family for flood frequency analysis and whether this practice can be improved through simple extensions of their methodology. Here, we propose the Köppen climate classification because it includes several of the variables that affect peak flow distributions (see the introduction - temperature, precipitation, vegetation, soil properties), and precipitation intensity because it represents aspects of flood-generating precipitation regimes (Hayden, 1988). Especially the Köppen classification forms a clear and relatively easy-to-apply selection criteria for probability distribution – as water managers would not be dependent on data. We have tried to make this clearer in the introduction in lines 99 to 101. To adhere to the necessity of making the distribution family widely applicable, it is relevant to also include records affected by human interferences. It implies that we make the assumption that over the recorded time the frequency distribution can change, but the distribution family will not.*

*Of course, this does not imply that urbanization and flood measures aren't relevant to the analysis and should be checked out. Instead of using the suggested GAGESII dataset, we tried a different method by using the peak streamflow qualification codes 5 and 6, indicating some degree of influence of "Regulation or Diversion." This provides a convenient check to see the effect of urbanization as we already had this information available through our own dataset. We first removed all records that contain data points assigned with the 5/6 code from the >30-year-long dataset (containing ~4200 records) and then selected the longest record for all remaining HUC regions (1017), as described in the method section. The L-moment diagrams from these analyses do not show major differences compared to the analyses that include regulated streams. Only records for continental regions do not follow the GEV distribution as significantly; however, the pattern remains. It should be mentioned here that the size of the Arid sample reduces by half. The results of these analyses were added to the Supporting Information (SI), and we chose to stick to the original results because of the reasons mentioned above.*

*We would like to stress that we believe any scientific analysis on individual records should take into account the non-stationarities caused by streamflow regulations. Here, we are mostly concerned with the first step of such an analysis, the recommended family distribution.*

*We did not remove years or records that contained codes, as they only represented a very small proportion of the total sample (less than 0.01% for all years in the case of code 4 and 8).*

A few additional methodological suggestions are:

1. The major impediment to making these results actionable is that no attempt is made to determine whether it is preferable to log-transform the flood peaks or not. To address this question, among other possible considerations, it seems that a primary need is the calculation of a goodness-of-fit

measure; for example, chapter 5 of Hosking and Wallis (1997) discusses one such measure. I think this matter should at least be acknowledged in the paper.

*We agree with Reviewer 4 that it is valuable to address this in the paper. We decided to use the sum of squared error (SSE) between the WMA and the theoretical distribution line of the GEV, LN3, and P3 distributions as a goodness-of-fit parameter. These results are presented in Table 2 and are mentioned throughout the results section.*

2. In response to reviewer 1, the authors suggested they would add an analysis of the effect of catchment area as an appendix, but apparently they did not. Based on the discussion of this basin property in the introduction, which agrees with my thinking, I think these results should indeed be added.

*We agree with Reviewer 4 and Reviewer 1 and will include these results in the SI in the Zenodo repository.*

3. Regarding the issue of regions that are rather large and likely heterogeneous, I don't feel that refining regions is a crucial issue, but I would point out that Hosking and Wallis (1997, ch. 4) provides L-moment-based methods for testing the homogeneity of regions. A related comment is that I'm not sure the inclusion of Alaska and Hawaii is helpful: they don't have many gauges and have rather different climates than areas of the conterminous United States.

*We believe this point relates to line 191 of the manuscript. This comment refers to use of a national recommended family distributions and in this case the entire United States. We agree that for regional analyses Hosking and Wallis (1997) provides a very good method to test homogeneity. The gauges of Hawaii and Alaska are not included in the Koppen analyses, because there are to little gages that belong to these particular groups (tropical and polar climates).*

4. As pointed out by Reviewer 2, consideration of PILFs is a powerful tool for focusing on the upper tail of flood distributions and could presumably be applied to other distributions in addition to LP3 (though it never has been to my knowledge). With the current focus on the complete flood distribution, avoiding its use is acceptable in my opinion. (But on the other hand, for non-extreme floods, fitting to a parametric distribution isn't needed for at-site flood frequency as interpolation could be used.)

5. Determination of L-moments and L-moment ratios implies determination of distribution parameters. Are these sensible? (For example, for non-log-transformed data, are location coefficients non-negative?)

*We included histograms of the mean, standard deviation and skew of the records in the SI (Fig. S4). We believe there are no problematic values.*

Beyond these methodological concerns and suggestions, I have several concerns regarding the presentation, mostly due to incompleteness:

1. Dataset table in data archive:
a. Provide definitions / units for the columns of dataset.
b. Limit table to stations actually used in the analysis.

*We agree with Reviewer 4 that this will make the dataset clearer and we will adjust this accordingly.*

2. In introduction:
a. Address quantile-dependent effects of urbanization on peaks (see for example, Konrad, 2003, and Over et al., 2016, and references therein).

*We agree with Reviewer 4 that this is valuable to mention and have included it in the text in line 61 and 62.*

b. Address effects of reservoirs (see for example FitzHugh and Vogel paper cited above and references therein).

*We agree with Reviewer 4 that this is valuable to mention and have included it in the text in line 64.*

c. Here or in the discussion, address the issue of possible trends in the flood-peak data used.

*We agree with Reviewer 4 that this is an important point, however the study addresses the selection of a distribution family, not the change in the individual peak flow distribution.*

3. Data section
a. Using the longest record for "each independent USGS hydrologic unit" to avoid bias toward heavily sampled rivers (lines 115-6) sounds reasonable, but what is an "independent USGS hydrologic unit"?

*Hydrologic units are used by the USGS to classify (sub)watersheds. We removed the word independent in the text as that perhaps explains the confusion.*

b. Suggest comparing the Koppen climatology to the flood climatology of Hayden (1988).

*We agree with Reviewer 4 that the Hayden (1988) flood climatology shows overlap with points we mention in our discussion. We could not find a digitized version of the map in Hayden (1988) meaning it is difficult to include it in the analysis. However the descriptions fit the conclusions we draw on the family distribution characteristics as described in the discussion and we acknowledge Hayden (1988) in the introduction (line 102) .*

c. The Koppen climate and Psc values were determined for each record were said to have been determined "by proximity" (line 126), which is quite vague. The proximity of what to what? Please explain more completely. If based on the location of the streamgauge, the authors should consider that the climate at the streamgauge location may be significantly different than the climate experienced by the watershed as a whole, depending on the size and other properties such as elevation range of the watershed. One easy modification would be to use the properties at the basin centroid, the location of which is given in the GAGES-II dataset cited above.

*Proximity was indeed based on the location of the stream gauge to the closed grid cell in the Koppen and precipitation dataset. We agree with the concern of Reviewer 4 that especially for large watersheds peak flows can be influenced by weather upstream of the gauge. We used the solution provided by Reviewer 4 to replace the coordinates of the gages with the basin centroids of GagesII (line 130). The results did not change meaningfully as for most gages there coordinates the stream gauge is closely located to the centroid.*

4. Presentation of L-moment analysis (Section. 2.2):
a. Define L-moment ratios L-Cs and L-Ks in terms of underlying L-moments.

*L-Cs and L-Ck are the L-moment ratio for skew and L-moment ratio for kurtosis. We agree with Reviewer 4 that the abbreviations and description were unclear so we changed it accordingly in line 162 to 165 to also better match the figures and Hosking and Wallis (1997).*

b. Variables in table 1 are not defined or are defined poorly (Notes at bottom have several typos).

*We agree with Reviewer 4 that the table caption and notes needed improvement. We updated the table, changed the notes to better describe the parameters, took out the typos, and referred better to Hosking and Wallis (1997).*

c. How were sample L-moments determined? (Note that there are small-sample biases in the "simple" versions: Hosking and Wallis, 1997, section 2.7.)

*We do use the simple version of as described in section 2.7, however most of our samples are (much) longer than 20 measurements (all are longer than 30 years) and the $t_4$ (L-Kurtosis) values are smaller than 0.4 (93%) – which are less affected by the bias according to Hosking and Wallis (1997).*

d. For the fits to log-transformed data, what did you do about the real-space values less than 1 whose logs are negative? And what was done with zero-valued peaks?

*We recoded zero-values to the lowest non-zero value in that record to be able to compute the log's. These records do not significantly influence the outcome of our results. For analyses with individual record in log-space we recommend censoring PILF's as described in Bulletin 17C.*

5. Accuracy of claims made:
Two places there are statements about showing that (or whether) hydroclimatic data can improve extreme flood probability estimates (lines 271-2, where this is stated as the main objective of the study), and lines 354-5, where it is stated that "probability model selection can be improved when it is based on the hydroclimatic properties of the basin". However, strictly, I don't think this was done, as no improvement in goodness of fit was provided (compare methodological suggestion 1.). It would be more accurate to say that it was shown that model selection can be guided by the use of hydroclimatic classification or similar language.

*We agree with Reviewer 4 that in the previous manuscript these statements were too strong. As we now included a goodness-of-fit measure we believe these statements are justified.*

I also added some editorial comments to a copy of the manuscript, which is attached.

*We adopted the editorial comments in the final manuscript. There were two comments that we wanted to provide with a reply:*

What is this? And why "also"? In addition to what? Explain more fully or if not important, start off the paragraph differently.

*Here we refer to the fact that the distribution of L-moment ratios within the L-moment diagram is dependent on local precipitation intensities, just as it was dependent on the Koppen climate classification. We adjusted this sentence to better reflect this.*

1. Is this is a general fact for any dataset or did you check it in your dataset? If the former, give a reference; if the latter, give details on how you checked this.

2. "the extreme flood" is somewhat odd, as if it is a technical term that everybody uses, but it is not, as far as I know. Either "extreme floods" or "the most extreme flood in each record" or something else if those don't convey your intended meaning.

*The purpose of this sentence is to practically describe what the variance of L-Skew means (and related that to the distribution lines). Here we refer to the fact that distributions with a relatively large skew have relatively thick tails. This means that extreme values are more extreme than the average compared to for example Gaussian distributions (with no skew). Accordingly 'the extreme flood' is not meant as a technical term. We agree with Reviewer 4 that 'extreme floods' would be a more appropriate term.*

References cited:

Falcone, J., 2011, GAGES-II: Geospatial Attributes of Gages for Evaluating Streamflow, https://water.usgs.gov/lookup/getspatial?gagesII_Sept2011, https://doi.org/10.3133/70046617.

FitzHugh, T.W. and Vogel, R.M. (2011), The impact of dams on flood flows in the United States. River Res. Applic., 27: 1192-1215. https://doi.org/10.1002/rra.1417.

Hayden, B.P., Flood climates, in Flood Geomorphology, edited by V.R. Baker, R.C. Kochel and P.C. Patton, pp. 13-26, John Wiley and Sons, New York. 1988.

Konrad, C.P., 2003, Effects of urban development on floods: U.S. Geological Survey Fact Sheet FS-076–03, 4 p., https://pubs.usgs.gov/fs/fs07603/.

Over, T.M., Saito, R.J., and Soong, D.T., 2016, Adjusting annual maximum peak discharges at selected stations in northeastern Illinois for changes in land-use conditions: U.S. Geological Survey Scientific Investigations Report 2016–5049, 33 p., https://doi.org/10.3133/sir20165049.